# Metabolomics, Genetics, and Environmental Factors: Intersecting Paths in Abdominal Aortic Aneurysm

**DOI:** 10.3390/ijms26041498

**Published:** 2025-02-11

**Authors:** Lilian Fernandes Silva, Jagadish Vangipurapu, Anniina Oravilahti, Aldons Jake Lusis, Markku Laakso

**Affiliations:** 1Institute of Clinical Medicine, Internal Medicine, University of Eastern Finland, 70210 Kuopio, Finland; lilianf@uef.fi (L.F.S.); jagadish.vangipurapu@uef.fi (J.V.); anniina.oravilahti@uef.fi (A.O.); 2Division of Cardiology, Department of Medicine, David Geffen School of Medicine, University of California, Los Angeles, CA 90095, USA; jlusis@mednet.ucla.edu; 3A.I. Virtanen Institute for Molecular Sciences, University of Eastern Finland, 70211 Kuopio, Finland; 4Department of Medicine, Kuopio University Hospital, 70200 Kuopio, Finland

**Keywords:** abdominal aortic aneurysm, metabolomics, genetics, xenobiotics

## Abstract

Abdominal aortic aneurysm represents a significant public health concern, particularly in men aged 55 to 64, where it occurs in about 1%. We investigated the metabolomics and genetics of AAA by analyzing a cohort including 76 patients with AAA and randomly selected 228 controls. Utilizing the Metabolon DiscoveryHD4 platform for non-targeted metabolomics profiling, we identified several novel metabolites significantly associated with AAA. These metabolites were primarily related to environmental and lifestyle factors, notably smoking and pesticide exposure, which underscores the influence of external factors on the progression of AAA. Additionally, several genetic variants were associated with xenobiotics, highlighting a genetic predisposition that may exacerbate the effects of these environmental exposures. The integration of metabolomic and genetic data provides compelling evidence that lifestyle, environmental, and genetic factors are intricately linked to the etiology of AAA. The results of our study not only deepen the understanding of the complex pathophysiology of AAA but also pave the way for the development of targeted therapeutic strategies.

## 1. Introduction

Abdominal aortic aneurysm (AAA), characterized by an aortic diameter exceeding 3 cm, constitutes a substantial global healthcare challenge [1]. The risk of AAA escalates notably after 60 years old. Clinically relevant aneurysms, exceeding 4 cm in diameter, are present in approximately 1% of men aged 55 to 64, with prevalence escalating by 2% to 4% per subsequent decade [2,3]. AAAs manifest four to six times more frequently in men than in women [4,5] and are more prevalent in white individuals compared to Black individuals [6]. AAA is characterized by localized structural deterioration of the wall of the aorta, resulting in progressive dilation and rupture [7]. AAA pathogenesis is closely related to the progressive depletion and dysfunction of vascular smooth muscle cells and includes proteolysis, oxidative stress, inflammatory immune response, and apoptosis [8]. These processes cause the loss of elasticity and resistance of the artery wall.

Smoking and hypertension are significant risk factors for AAA. Additionally, AAAs are more prevalent in individuals with atherosclerosis, with an approximate 5% prevalence in those with coronary artery disease [9,10]. Positive family history substantially increases the risk of AAA [11,12], suggesting that genetic factors play an important role in the development of AAA [13]. AAA appears less common in individuals with diabetes [6].

The core of AAA management requires longitudinal surveillance until the aneurysm reaches a size where the risk of rupture surpasses the risk of repair [14]. Inflammatory processes [15,16] and gut microbiota [17] have important roles in the pathogenesis of AAA. A recent study based on 449,463 participants from the UK Biobank reported that exposure to long-term air pollutants increased the risk of AAA [18]. Similarly, exposure to environmental toxins, polycyclic aromatic hydrocarbons (PAH), has been implicated in cardiovascular diseases and AAA [19].

In recent years there has been a growing interest in exploring the metabolic and genetic profiles associated with AAA to understand its multifactorial nature. Previous metabolomics studies on AAA have been limited by small sample sizes and a small number of metabolites analyzed, leading to inconsistent results [20,21,22,23,24,25,26,27,28,29]. Metabolites are crucial for cellular functions, influencing various physiological processes and signaling pathways. Given the several risk factors and metabolic pathways affected in patients with AAA, the metabolomics approach is likely to identify pathological processes in AAA expansion and allow finding novel therapeutic strategies.

The identification of circulating biomarkers with diagnostic and prognostic values for the diagnosis of AAA is challenging. A biomarker is a measurable indicator of a biological state or condition that is used to assess health or disease status or to evaluate responses to a therapeutic intervention. Based on the pathophysiology of an AAA, circulating biomarkers can be classified according to their relationship with prothrombotic activity, degradation of the extracellular matrix of the vascular wall, or the immunoinflammatory. To comprehend the impact of the genetic background and associated pathways in the development of AAA, we applied a comprehensive high-throughput liquid chromatography–tandem mass spectroscopy (LC-MS/MS) in the patients with AAA compared to matched controls in the Finnish population-based METSIM study.

## 2. Results

### 2.1. Baseline Characteristics of AAA Cases and Controls

Table 1 shows the baseline characteristics of the participants with AAA and randomly selected controls. Participants with AAA were older, had higher systolic blood pressure, higher concentration of triglycerides and hs-CRP, lower insulin sensitivity (Matsuda), and a higher percentage of smokers, and statin treatment.

### 2.2. Differences in the Metabolite Abundances Between the AAA Cases and Controls

We found statistically significant differences in 25 metabolites between the participants with AAA and the controls. The participants with AAA had an increased abundance of xenobiotics (n = 10), a co-factor, beta-cryptoxanthin (n = 1), a carbohydrate N-acetylneuraminate (n = 1), 4 amino acids, and 9 lipids when compared to the controls (Table 2). The most significant differences between the cases and the controls among the xenobiotics were for 2-naphthol sulfate (*p* = 3.5 × 10^−9^), methyl naphthyl sulfate (*p* = 1.7 × 10^−7^), and N-(2-furoyl) glycine (*p* = 1.1 × 10^−6^). Additionally, we found that abundances of a xenobiotic (piperine), a co-factor (β-cryptoxanthin), a lactosylceramide (lactosyl-N-palmitoyl-sphingosine (d18:1/16:0)), and two lysoplasmalogens (1-linoleoyl-GPC (18:2) and 1-stearoyl-GPC (18:0)), were decreased in the cases compared to the controls. After the adjustment for statin medication all xenobiotics, excluding O-cresol sulfate, remain statistically significant as well as beta-cryptoxanthin, N-acetylneuraminate, and four amino acids. Among the nine lipids only 2-hydroxyarachidate remained statistically significant.

Appendix A presents the metabolites nominally associated with AAA (*p* < 0.05). We identified 301 metabolites, including 163 positively and 138 negatively associated with AAA, incluiding lipids (n = 135), amino acids (n = 66), and xenobiotics (n = 44). Hippurate, bilirubin, biliverdin, phosphatidylcholine, lysophosphatidylcholine, aspartate, glutamate, and glutamine were nominally associated with AAA. Similar findings have been reported in previous small studies AAA [21,23,24].

### 2.3. Correlation Between the Metabolites Significantly Associated with AAA After Adjustment

Figure 1 illustrates the intercorrelations among 16 metabolites. We observed positive correlations involving N-(2-furoyl)glycine and 3-methylcatechol sulfate, both of which are implicated in detoxification processes. Similarly, lanthionine and C-glycosyltryptophan demonstrated a positive correlation. Further, there were positive correlations among 3-hydroxy-3-methylglutarate, 2-naphthol sulfate, and methylnaphthyl sulfate. Additionally, 3-methylcatechol sulfate, N-(2-furoyl)glycine, and (2,4 or 2,5)-dimethylphenol sulfate also displayed positive associations. Positive relationships were also found between methylnaphthyl sulfate and C-glycosyltryptophan, as well as between 4-vinylphenol sulfate and 2-naphthol sulfate.

Inverse correlations were identified between piperine and all other 15 metabolites analyzed. Beta-cryptoxanthin was inversely associated with the same metabolites as piperine. Additionally, inverse correlations were observed between 3-hydroxy-3-methylglutarate and both 2-naphthol sulfate and methylnaphthyl sulfate. Although positively correlated among themselves, these metabolites exhibited inverse relationships with the ketone body metabolite 3-hydroxy-3-methylglutarate, suggesting a complex interplay between energy production and detoxification mechanisms. Methylnaphthyl sulfate and lanthionine also showed inverse correlations.

### 2.4. Association of Genetic Variants with Metabolites in Patients with AAA

We utilized the GWAS database “https://www.ebi.ac.uk/gwas/ (accessed on 13 November 2024)” to identify the genetic variants and the genes significantly associated with metabolites in the participants with AAA. Table 3 presents the associations at the genome-wide significance level (*p* < 5 × 10^−8^). For each gene, the genetic variant having the most significant association with the metabolites was selected. Six of the ten xenobiotics were significantly associated with genetic variants, and four were not associated with any genetic variants (methylnapht sulfafe (2), N-(2-furoyl)glycine, 4-vinylcatechol sulfate, (2,4 or 2,5)-dimethylphenol sulfate). Two other metabolites (lipid 2-hydroxyarachidate and amino acid lanthionine) were not associated with any genetic variant). Additionally, three metabolites were associated only with one genetic variant (2-naphthol sulfate, piperine, o-cresol sulfate).

A variant of rs169828 of the *ARSL* gene was significantly associated with 2-naphthol sulfate (*p* = 2 × 10^−28^). The *ARSL* gene is a sulfatase and makes an enzyme arylsulfatase. The function of this enzyme is not known [30]. O-cresol sulfate was associated with a variant of the *SGF29* gene (rs480400, *p* = 8 × 10^−12^). *SGF29* specifically recognizes and binds methylated “Lys-4” of histone H3, and non-histone proteins that are methylated on Lys residues [31]. 3-ethylcatechol sulfate was associated with the genetic variant of *SLC51A* (rs6795511, *p* = 1 × 10^−11^).

Piperine was associated with rs8041357 variant of the *ARID3B* gene. Gene Ontology annotations related to this gene include RNA polymerase II cis-regulatory region sequence-specific DNA binding and RNA polymerase II cis-regulatory region sequence-specific DNA binding [32]. Members of the *ARID* family have roles in embryonic patterning, cell lineage gene regulation, cell cycle control, transcriptional regulation, and possibly in chromatin structure modification [33]. N-acetylneuraminate was associated with a genetic variant in *ARHGEF3* (rs1354034, *p* = 4 × 10^−78^), which regulates skeletal muscle regeneration.

Other significant findings include an association of 1-(1-enyl-palmitoyl)-2-linoleoyl-GPC (P-16:0/18:2) with rs56228609-T of the *HERPUD1* gene (*p* = 1 × 10^−25^). This gene may play a role in unfolded protein response (UPR) and endoplasmic reticulum-associated protein degradation (ERAD). The *HERPUD1* gene expression is induced by UPR, and it has an ER stress response element in its promoter region, while the encoded protein has an N-terminal ubiquitin-like domain, which may interact with the ERAD system. This protein has been shown to interact with presenilin proteins and increase the level of amyloid-beta protein following its overexpression [32]. β-cryptoxanthin was associated with a genetic variant, rs75226183 in *RNUE-54P* pseudogene. β-cryptoxanthin is known for its antioxidant properties [34].

## 3. Discussion

Our study integrates metabolomics and genetics to explore potential biomarkers associated with AAA in the METSIM cohort. Our study reports several novel findings. We found that among the 25 metabolites identified, 16 metabolites remained statistically significant after the adjustment for statin medication. Among them, nine xenobiotics were significantly associated with AAA.

Xenobiotics are chemical substances including plants, drugs, pesticides, food additives, chemicals, and environmental pollutants [35]. We observed that the abundance of xenobiotics was significantly increased in the participants with AAA compared to the controls, suggesting that an altered xenobiotic metabolism contributes to the pathophysiological processes resulting in aneurysm formation. Our findings indicate that exposure to environmental toxins, including PAHs, pesticides, and herbicides, plays a major role in the development of AAA [19,36,37].

Three metabolites positively associated with AAA in our study have been previously associated with tobacco smoking (2-naphthol sulfate, methylnaphthyl sulfate, and 4-vinylphenol sulfate) [38]. Exposure to PAHs increases the risk of cardiovascular diseases, including AAA [19,39]. PAH compounds are organic compounds found in tobacco and tobacco smoke, formed primarily during the incomplete combustion of organic materials [40,41,42,43]. PAHs induce endothelial dysfunction, oxidative stress, and inflammation, which weaken the aortic wall, resulting in vascular remodeling, arterial stiffness, and plaque formation [40,41,42,43]. Chronic PAH exposure increases the risk of AAA, especially in individuals having several risk factors for AAA.

We found one metabolite is a derivative of pesticides/herbicides, (2,4 or 2,5)-dimethylphenol sulfate. Pesticides and herbicides increase the risk of AAA by causing oxidative stress, inflammation, and arterial stiffness by degrading structural proteins, such as elastin and collagen, which are important for the integrity of the aorta [36,37]. Additionally, pesticides disrupt normal metabolic and immune processes that regulate vascular tissue remodeling, particularly in individuals with several risk factors for AAA [36,37].

The significant metabolite differences we observed between the AAA cases and controls, particularly in the xenobiotics class, suggest a potential dysregulation in the body’s ability to process and eliminate these compounds in patients with AAA. Increased abundances of 2-naphthol sulfate, methylnaphthyl sulfate, and 4-vinylphenol sulfate indicate an altered xenobiotic metabolism, which could contribute to pathophysiological processes leading to aneurysm formation and progression. These findings support the hypothesis that exposure to environmental toxins increases the risk of AAA, as suggested by the studies linking cardiovascular diseases and environmental pollutants PAHs, pesticides, and herbicides to an increased risk of AAA [19,36,37].

N-(2-furoyl)glycine is a metabolite generated by microbiota and found in food prepared by strong heat. This metabolite belongs to the class of N-acyl-alpha amino acids and is a product of fatty acid catabolism and regulates mitochondrial fatty acid beta-oxidation [44]. N-(2-furoyl)glycine participates in the pathways increasing oxidative stress, inflammation, and mitochondrial dysfunction, which are risk factors for cardiovascular diseases [45]. We found an association between the carbohydrate conjugate N-acetyl-alpha-neuraminate, a sialic acid found on the surface of various cell types, and a positive association with AAA. Sialic acids play a vital role in mediating cell–cell and cell–molecule interactions in eukaryotes, and they can be used by pathogens like *E. coli* to evade host immune responses [46]. Infections of bacterial and fungal origin are known to contribute to the development of infectious AAA, which is associated with an elevated risk of aneurysm rupture [47].

Abundance of piperine was found to be decreased in the participants with AAA. Piperine has many pharmacological effects and several health benefits, especially against chronic diseases, such as increased insulin sensitivity, anti-inflammatory effects, and improvement of hepatic steatosis [48]. Piperine has been shown to attenuate pathological cardiac fibrosis via PPAR-γ/AKT pathways [49]. Inflammatory abdominal aortic aneurysm is characterized by extensive fibrosis, thickened walls, and dense adhesions observed in 3–10% of all cases of AAAs. Surgery is technically challenging and is associated with increased morbidity and mortality [50]. We found an increased abundance of piperidine in the controls, suggesting a protective effect against AAA.

We performed a correlation analysis between the 16 metabolites (Figure 1). We identified positive correlations among 3-hydroxy-3-methylglutarate, 2-naphthol sulfate, and methylnaphthyl sulfate, which play significant roles in energy metabolism and detoxification. This suggests intricate interactions in metabolic pathways crucial for breaking down harmful compounds. Additionally, we observed important relationships within detoxification pathways involving 3-methylcatechol sulfate, N-(2-furoyl)glycine, and dimethylphenol sulfate, essential for managing oxidative stress and inflammation. We also noted inverse correlations between piperine, beta-cryptoxanthin, and metabolites related to amino acid metabolism and detoxification. These findings indicate that piperine and beta-cryptoxanthin might influence nutrient absorption and metabolism, pointing to potential impacts on metabolic health.

We identified significant genetic associations with metabolites linked to AAA. Key findings include the association of the *ARSL* gene variant rs169828 with 2-naphthol sulfate. Additionally, the variants in the *SLC51A*, *ARID3B*, and *ARHGEF3* genes were associated with 3-ethylcatechol sulfate, piperine, and N-acetylneuraminate. Additionally, the pseudogene *RNUE-54P* was linked to β-cryptoxanthin, known for its antioxidant properties [34]. These findings not only enhance our understanding of the genetic factors influencing metabolite variations associated with AAA but also point to specific molecular pathways that may be targeted for therapeutic intervention, particularly the interaction between protective lipid molecules and protein response mechanisms.

Previous studies have reported associations of several metabolites with AAA, including hippurate, biliverdin, bilirubin, proline, glycerol, aspartate, glutamate, glutamine, proline, citric acid, 2-oxoglutaric acid, succinic acid, phosphatidylcholines, and lysophosphatidylcholines [20,21,22,23,24,25,26,27,28,29]. However, these findings have not been consistent in other studies. In our study, using the nominally significant threshold (*p* < 0.05), we were able to replicate most of these findings, including the associations with hippurate, biliverdin, bilirubin, phosphatidylcholine, lysophosphatidylcholines, aspartate, glutamate, and glutamine. This suggests that increasing the sample size is essential to identify new metabolites significantly associated with AAA.

Figure 2 summarizes our findings and underscores the critical links between xenobiotics in the development of AAA, influenced by lifestyle, environmental toxins, and genetic factors. We identified key metabolites and genetic variants contributing to the pathophysiology of AAA.

In summary, our study provides a comprehensive analysis of the metabolomics and genetic risk factors associated with AAA in the METSIM cohort, offering new insights into the pathophysiology of AAA. We identified eight xenobiotics positively associated with AAA, highlighting the role of environmental toxins such as PAHs, pesticides, and herbicides in increasing vascular remodeling, oxidative stress, and inflammation (Appendix A). Genetic analyses revealed associations between the key metabolites, and specific genetic variants, emphasizing an independent effect of genetic predisposition on the risk of AAA. The preventive role of the metabolites, especially piperine and beta-cryptoxanthin, suggests potential therapeutic avenues.

The limitations of our study are that it included only middle-aged and elderly Finnish men, which limits the generalizability of our findings to women and other populations. Additionally, the use of cross-sectional data restricts our ability to establish causality between metabolic factors and AAA. Future studies are needed to investigate metabolomics and genetics in larger, more diverse populations to validate our findings in the pathophysiology and etiology of AAA. Longitudinal studies tracking metabolic changes over time in individuals at high risk for AAA may help to identify early biomarkers, providing an opportunity for preventive interventions. Given the strong associations between xenobiotics and AAA, studies into environmental and lifestyle modifications could offer practical applications to reduce the incidence of AAA. Further exploration of the metabolic pathways involved in xenobiotic metabolism may lead to the development of targeted therapies to mitigate harmful effects of environmental toxins. The identification of potential biomarkers also opens avenues for precision medicine approaches, where metabolic profiling could guide personalized monitoring and treatment strategies for individuals at high risk of AAA.

## 4. Materials and Methods

### 4.1. Study Population

The METSIM study includes 10,197 men, aged from 45 to 73 years at baseline, and randomly selected from the population register of Kuopio, Eastern Finland. The METSIM study was approved by the Ethics Committee at the Kuopio University Hospital, Finland. All participants provided written informed consent. The design and methods of the METSIM study have been previously described in detail [51,52]. A total of 304 men from the METSIM study were included in the current study, 76 participants having AAA and 228 random controls.

### 4.2. Clinical and Laboratory Measurements

Height was measured without shoes to the nearest 0.5 cm. Weight was measured in light clothing with a calibrated digital scale (Seca 877, Hamburg, Germany). Laboratory studies after 12 h fasting included the following measurements: plasma glucose and insulin, lipids, lipoproteins, and mass spectrometry metabolomics (Metabolon, Durham, NC, USA). An oral glucose tolerance test was performed to evaluate glucose tolerance (75 g of glucose). Clinical and laboratory measurement methods have been previously published [41]. Briefly, plasma glucose was measured by enzymatic hexokinase photometric assay (Konelab Systems Reagents, Thermo Fischer Scientific, Vantaa, Finland). Insulin was determined by immunoassay (ADVIA Centaur Insulin IRI, no. 02230141, Siemens Medical Solutions Diagnostics, Tarrytown, NY, USA). Serum alanine aminotransferase (ALT) was measured by an enzymatic photometric test (Konelab Reagent System, Thermo Fisher Scientific, Vantaa, Finland). BMI was calculated as weight divided by height squared. Smoking status was defined as current smoking. Other laboratory measurements have been previously reported [51]. Estimated glomerular filtration rate (eGFR) was calculated using the CKD-Epi equation [52].

### 4.3. Metabolomics

Metabolites were measured by using Metabolon Inc.’s untargeted Discovery HD4 platform based on ultra-high-performance liquid chromatography–tandem mass spectroscopy (UPLC-MS/MS) (Metabolon, Morrisville, NC, USA). Samples stored at −80 °C prior to analysis were prepared using the automated MicroLab STAR^®^ system (Hamilton Company, Reno, NV, USA). Several recovery standards were added prior to the first step in the extraction process for quality control (QC) purposes. A pooled matrix sample generated by taking a small volume of each experimental sample served as a technical replicate throughout the dataset. Extracted water samples served as process blanks, and QC standards that were carefully chosen not to interfere with the measurement of endogenous compounds were spiked into every analyzed sample, allowing instrument performance monitoring and aiding chromatographic alignment. Overall process variability was determined by calculating the median relative standard deviation for all endogenous metabolites present in 100% of the pooled matrix samples. A data normalization step was performed to correct variation resulting from instrument inter-day tuning differences in studies spanning multiple days. Experimental samples were randomized across the platform run with QC samples spaced evenly. Raw data were extracted, peak-identified, and QC processed using Metabolon DiscoveryHD4 platform, and peaks were quantified using area under the curve. Compounds were identified by comparison to library entries of purified standards or recurrent unknown entities. Library matches for each compound were checked for each sample and corrected if necessary. Each metabolite was rescaled to set the median equal to 1.

The Metabolon DiscoveryHD4 platform identified a total of 1540 metabolites. From this initial set, only metabolites with at least 40% complete data across the dataset were retained, while all metabolites lacking identification information were excluded, resulting in 1009 metabolites for statistical analysis. All samples were processed together for peak quantification and data scaling. Evaluation of overall process variability by the median relative standard deviation for endogenous quantified raw mass spectrometry peaks for each metabolite using the area under the curve and metabolites present in all 20 technical replicates in each batch was performed. Variation was adjusted for day-to-day instrument tuning differences and columns used for biochemical extraction by scaling the raw peak quantifications to the median for each metabolite by the Metabolon batch.

### 4.4. Selection of Genetic Variants Associated with AAA

We identified genetic variants associated with AAA from previous publications and the GWAS Catalog (The NHGRI-EBI Catalog of human genome-wide association studies) “https://www.ebi.ac.uk/gwas/ (accessed on 13 November 2024)” in individuals of European ancestry. Among the genetic variants for each gene, we selected the variant having the most significant association with AAA.

### 4.5. Statistical Analysis

We conducted statistical analyses using IBM SPSS Statistics, version 29. We log-transformed all continuous variables except for age to correct for their skewed distribution. We used one-way ANOVA to assess the differences in clinical traits and metabolites between the two groups, participants with AAA and controls. We used Bonferroni correction to define the statistical significance threshold. The threshold for statistical significance was ≤5.0 × 10^−5^, given the 1009 metabolites included in the statistical analysis. Correlations between the metabolites were calculated using the Pearson correlation. The correlation figure was generated using the Python programming language.

## 5. Conclusions

We applied metabolomics and genetics to identify novel metabolites associated with AAA in the METSIM cohort. Among the 16 metabolites significantly associated with AAA, 10 were xenobiotics linked to lifestyle and environmental exposures, including substances from benzoate metabolism, smoking, and pesticides. Genetic analyses highlighted several genetic variants associated with metabolite abundances positively associated with AAA. Our comprehensive analysis, combining metabolomics with genetic data, robustly demonstrates that lifestyle and environmental influences and genetic factors play significant roles in the etiology of AAA.

## Figures and Tables

**Figure 1 ijms-26-01498-f001:**
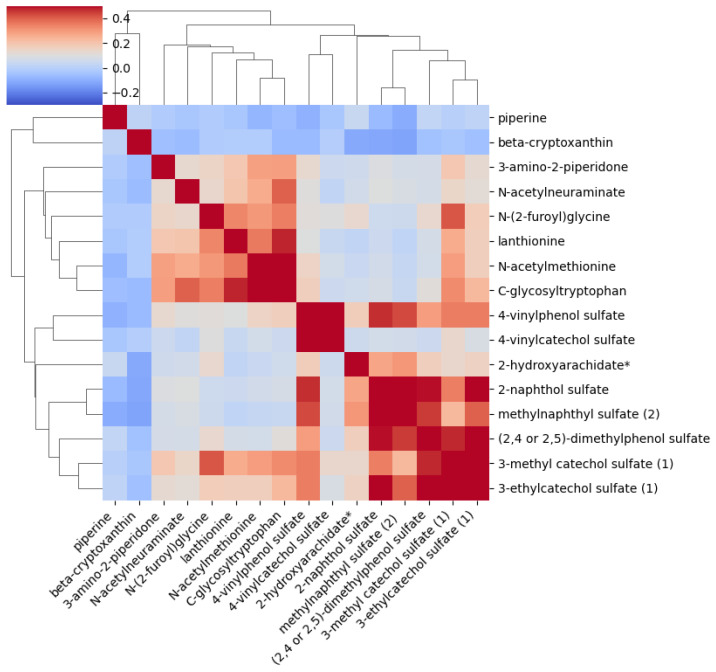
The heatmap illustrates the intercorrelations of 16 metabolites significantly associated with AAA. Positive correlations are shown in red and negative correlations in blue.

**Figure 2 ijms-26-01498-f002:**
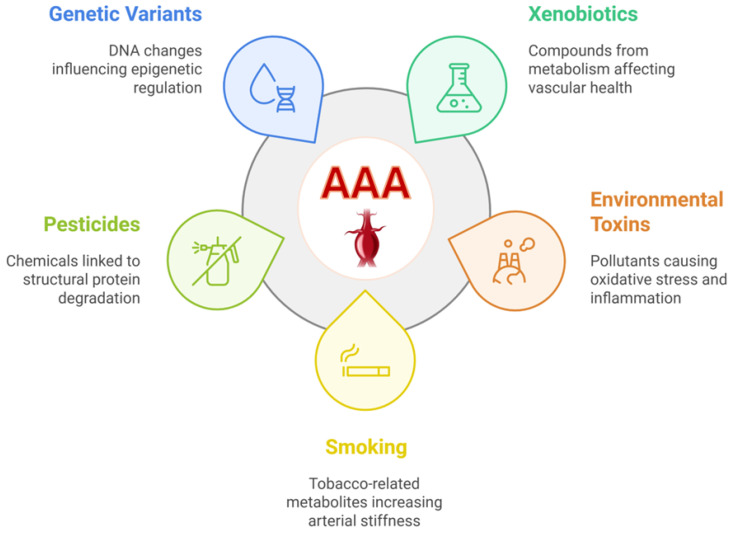
Overview of the contributing factors associated with the development of AAA. The figure illustrates the associations identified in the METSIM cohort, highlighting the interplay between lifestyle, environmental toxins, and genetic factors in the development of AAA.

**Table 1 ijms-26-01498-t001:** Baseline characteristics of AAA cases and controls.

Variable	Cases (n = 76)	Controls (n = 228)	*p*
	Mean ± SD	Mean ± SD	
Age (years)	62.91 ± 5.72	57.44 ± 7.24	6.0 × 10^−9^
Body mass index (kg/m^2^)	27.48 ± 4.29	26.83 ± 3.73	0.207
Systolic blood pressure (mmHg)	140.93 ± 15.48	136.02 ± 16.19	0.021
Total triglycerides (mmol/L)	1.71 ± 1.52	1.34 ± 0.69	0.004
LDL cholesterol (mmol/L)	3.17 ± 0.85	3.40 ± 1.00	0.070
Fasting plasma glucose (mmol/L)	5.75 ± 0.51	5.74 ± 0.48	0.882
Matsuda ISI (mg/dL, mU/L)	5.26 ± 3.08	6.81 ± 3.99	0.002
eGFR (mL/min/1.73 m^2^)	81.13 ± 16.25	87.80 ± 12.06	1.7 × 10^−4^
hs-CRP (mg/L)	4.03 ± 7.09	1.72 ± 1.68	7.2 × 10^−6^
Smoking (%)	40.8	14.5	3.6 × 10^−6^
Statin medication (%)	47.4	23.2	8.3 × 10^−5^

Abbreviations: eGFR, estimated glomerular filtration rate; hs-CRP, high sensitivity C-reactive protein, LDL, low-density lipoprotein. *p* values were calculated with ANOVA and the chi-square test. Controls were matched for age, BMI, and systolic blood pressure.

**Table 2 ijms-26-01498-t002:** Statistically significant differences in metabolite abundances between the participants with AAA and the controls.

Metabolite	Subclass	Cases		Controls		*p*	*p* *
		n	Mean ± SD	n	Mean ± SD		
**Xenobiotics**							
2-naphthol sulfate	Chemical	71	0.67 ± 1.13	217	−0.15 ± 0.93	**3.5 × 10^−9^**	**7.8 × 10^−9^**
Methylnaphthyl sulfate	Chemical	55	0.66 ± 1.11	125	−0.21 ± 0.94	**1.7 × 10^−7^**	**7.0 × 10^−8^**
N-(2-furoyl)glycine	Food Component/ Plant	63	0.53 ± 1.19	174	−0.18 ± 0.88	**1.1 × 10^−6^**	**1.8 × 10^−6^**
4-vinylphenol sulfate	Benzoate Metab.	76	0.60 ± 1.23	228	−0.08 ± 0.99	**1.4 × 10^−6^**	**6.4 × 10^−7^**
Piperine	Food Component/ Plant	74	−0.41 ± 1.20	225	0.19 ± 0.89	**5.8 × 10^−6^**	**3.9 × 10^−7^**
3-methyl catechol sulfate	Benzoate Metab.	76	0.55 ± 0.91	227	−0.07 ± 1.06	**8.8 × 10^−6^**	**3.2 × 10^−5^**
4-vinylcatechol sulfate	Benzoate Metab.	73	0.55 ± 1.14	217	−0.05 ± 0.99	**2.4 × 10^−5^**	**3.1 × 10^−5^**
O-cresol sulfate	Benzoate Metab.	70	0.58 ± 1.05	180	−0.04 ± 1.06	**4.2 × 10^−5^**	8.6 × 10^−5^
3-ethylcatechol sulfate	Food Component/ Plant	69	0.61 ± 0.95	195	−0.004 ± 1.02	**1.7 × 10^−5^**	**4.7 × 10^−5^**
(2,4 or 2,5)-dimethylphenol sulfate	Food Component/ Plant	62	0.52 ± 0.88	148	−0.12 ± 1.01	**2.6 × 10^−5^**	**2.6 × 10^−5^**
**Cofactors/vitamins**							
Beta-cryptoxanthin	Vitamin A Met.	73	−0.50 ± 1.04	225	0.09 ± 0.94	**7.6 × 10^−6^**	**1.8 × 10^−5^**
**Carbohydrate**							
N-acetyl neuraminate	Aminosugars	76	0.45 ± 1.04	228	−0.13 ± 0.87	**2.7 × 10^−6^**	**1.8 × 10^−6^**
**Amino acids**							
3-amino-2- piperidone	Urea cycle; Arginine and Proline Metab.	75	0.57 ± 1.04	228	−0.09 ± 0.95	**5.9 × 10^−7^**	**1.3 × 10^−6^**
N-acetyl methionine	Methionine, Cysteine, SAM and Taurine Metab.	76	0.52 ± 1.26	228	−0.12 ± 0.96	**6.0 × 10^−6^**	**9.5 × 10^−6^**
Lanthionine	Methionine, Cysteine, SAM and Taurine Metab.	55	0.65 ± 1.15	156	−0.03 ± 0.92	**1.4 × 10^−5^**	**2.3 × 10^−5^**
C-glycosyl tryptophan	Tryptophan Metab.	76	0.53 ± 1.34	228	−0.09 ± 0.95	**1.2 × 10^−5^**	**4.9 × 10^−6^**
**Lipid**							
2-hydroxy arachidate	Fatty Acid, Monohydroxy	73	0.49 ± 0.85	215	−0.07 ± 0.94	**8.2 × 10^−6^**	**9.1 × 10^−6^**
3-hydroxy-3-methylglutarate	Mevalonate Metab.	75	0.52 ± 1.11	228	−0.09 ± 0.96	**7.7 × 10^−6^**	0.002
Lactosyl-N-palmitoyl-sphingosine (d18:1/16:0)	Lactosylceramides	76	−0.39 ± 1.00	228	0.16 ± 1.01	**4.0 × 10^−5^**	0.002
1-linoleoyl-GPC (18:2)	Lysophospho lipid	76	−0.51 ± 1.12	228	0.09 ± 0.97	**7.1 × 10^−6^**	1.8 × 10^−4^
1-stearoyl-GPC (18:0)	Lysophospho lipid	76	−0.44 ± 1.09	228	0.16 ± 0.91	**3.4 × 10^−6^**	9.8 × 10^−5^
1-(1-enyl-palmitoyl)-GPC (P-16:0)	Lysoplasmalogen	76	−0.53 ± 1.15	228	0.06 ± 0.99	**2.0 × 10^−5^**	2.5 × 10^−4^
1-(1-enyl-palmitoyl)-2-linoleoyl-GPC (P-16:0/18:2)	Plasmalogen	76	−0.64 ± 1.06	228	0.03 ± 1.05	**2.8 × 10^−6^**	8.1 × 10^−5^
1-palmityl-2-palmitoyl-GPC (O-16:0/16:0)	Plasmalogen	75	−0.43 ± 1.05	228	0.10 ± 0.92	**2.8 × 10^−5^**	0.002
3beta,7alpha-dihydroxy-5-cholestenoate	Sterol	71	0.46 ± 1.03	217	−0.15 ± 1.04	**2.4 × 10^−5^**	5.1 × 10^−5^

Abbreviations: Metab., metabolism; *p* without adjustment; *p* * adjusted for the use of statin treatment; bold statistically significant.

**Table 3 ijms-26-01498-t003:** Metabolites associated with AAA and genetic variants at the genome-wide significance level.

Metabolite	Genetic Var.	*p* Value	Beta	Gene	No
**Xenobiotics**					
2-naphthol sulfate	rs169828-T	2 × 10^−28^	0.15	*ARSL*	1
Methylnapht sulfafe (2)	none	none	none	none	0
N-(2-furoyl)glycine	none	none	none	none	0
4-vinylphenol sulfate	rs8017367-?	2 × 10^−6^	0.23	*SOS2*	8
Piperine	rs8041357-T	8 × 10^−9^	0.17	*ARID3B*	1
3-methyl catechol sulfate	rs113759232-T	5 × 10^−13^	−0.50	*LINC02499*	5
3-ethylcatechol sulfate (1)	rs6795511-A	1 × 10^−11^	0.18	*SLC51A*	2
4-vinylcatechol sulfate	none	none	none	none	0
(2,4 or 2,5)-dimethylphenol sulfate	none	none	none	none	0
O-cresol sulfate	rs480400-G	8 × 10^−12^	−0.10	*SGF29*	1
**Lipids**					
1-(1-enyl-palmitoyl)-2-linoleoyl-GPC (P-16:0/18:2)	rs56228609-T	1 × 10^−25^	0.12	*HERPUD1*	6
1-stearoyl-GPC (18:0)	rs4665972-T	2 × 10^−21^	0.10	*SNX17*	11
1-linoleoyl-GPC (18:2)	rs174564-G	3 × 10^−24^	0.15	*FADS1*	9
3-hydroxy-3-methylglutarate	rs541190935	5 × 10^−10^	?	*CCDC177*	7
2-hydroxyarachidate	none	none	none	none	0
1-(1-enylpalmitoyl)-GPC (P-16:0)	rs3834458-?	2 × 10^−43^	−0.15	*FADS2*	12
3beta, 7alpha-dihydroxy-5-cholestenoate	rs1573558-T	3 × 10^−42^	0.26	*LINC02732*	2
1-palmityl-2-palmitoyl-GPC (O-16:0/16:0)	rs3764261-A	1 × 10^−16^	0.14	*HERPUD1*	3
Lactosyl-N-palmitoyl-sphingosine (d18:1/16:0)	rs4147929-A	7 × 10^−28^	−0.22	*ABCA7*	9
**Amino acids**					
3-amino-2-piperidone	rs121965043-G	4 × 10^−35^	0.91	*OAT*	3
N-acetylmethionine	rs6710438-C	1 × 10^−21^	0.19	*ALMS1*	19
C-glycosyltryptophan	rs34533854-C	3 × 10^−11^	0.09	*SHROOM3*	5
Lanthionine	none	none	none	none	0
**Carbohydrates**					
N-acetylneuraminate	rs1354034-T	1 × 10^−23^	-0.15	*ARHGEF3*	18
**Co-factors**					
Beta-cryptoxanthin	rs75226183-?	2 × 10^−9^	none	*RNUE-54P*	1

Abbreviations, Genetic Var, genetic variant; No, number of metabolites associated with the genetic variant.

## Data Availability

The data that support the findings of this study are available from the corresponding authors [M.L. and L.F.S.] upon reasonable request.

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
