# Peer review of "Metabolomics, Genetics, and Environmental Factors: Intersecting Paths in Abdominal Aortic Aneurysm"

_ijms, 2025, doi:10.3390/ijms26041498_

Round 1
Reviewer 1 Report
Comments and Suggestions for Authors
The authors have screened different metabolites in AAA patients in Finland and revealed both environmental and genetic risk factors for AAA development, the study is of clinical significance to identify potential new risk factors, biomarkers, and therapeutic targets. I have some minor suggestions and questions:
1. Biliverdin level is related to hemoglobin metabolism, which may reflect AAA progression, suggesting adding HB and platelet count in Table 1.
2. Is there any association between these metabolites and AAA size, ILT, or rupture outcomes?
3. The authors have nicely interpreted the deregulated metabolites and the lifestyle, environment, and related signaling pathways in the pathophysiological processes of AAA in the discussion, it’ll be clearer to combine the metabolites and genes with genetic variants into a pathway enrichment figure, or separate figures summarizing the related pathways.
Author Response
The authors have screened different metabolites in AAA patients in Finland and revealed both environmental and genetic risk factors for AAA development, the study is of clinical significance to identify potential new risk factors, biomarkers, and therapeutic targets. I have some minor suggestions and questions:
- Biliverdin level is related to hemoglobin metabolism, which may reflect AAA progression, suggesting adding HB and platelet count in Table 1.
Reply: The suggestion to include hemoglobin (HB) and platelet counts to investigate their correlation with biliverdin levels is insightful. However, as these data were not collected in the METSIM study we cannot provide these analyses in the current manuscript. Future studies could explore these parameters to enhance the understanding of the pathophysiological processes in AAA. Additionally, after the new statistical analysis suggested by Reviewer 2 , biliverdin was only nominally significant, and therefore we need more studies to clarify the role of biliverdin in AAA.
- Is there any association between these metabolites and AAA size, ILT, or rupture outcomes?
Reply: This manuscript focuses on the metabolic and genetic differences between AAA patients and controls. The association between these metabolites and clinical outcomes such as aneurysm size, intraluminal thrombus (ILT), or rupture were not available in our study due to the scope and design of the original data collection.
- The authors have nicely interpreted the deregulated metabolites and the lifestyle, environment, and related signaling pathways in the pathophysiological processes of AAA in the discussion, it’ll be clearer to combine the metabolites and genes with genetic variants into a pathway enrichment figure, or separate figures summarizing the related pathways.
Reply: Thank you for the suggestion to visually represent the interaction between metabolites and genetic variants through pathway enrichment figures. We agree that this could clarify the complex interactions, and we incorporated a separate pathway diagram in the Supplementary Information (Supplementary Figure 1) to illustrate these relationships in our revised manuscript.

Reviewer 2 Report
Comments and Suggestions for Authors
The manuscript submitted by Fernandes Silva et al presents an interesting and relevant study of metabolomics, genetic and environmental factors in patients with Abdominal Aortic Aneurysm (AAA), a life-threatening condition. The research topic is very relevant, but few topics need to be clarified.
For this study the authors used a subset of a male cohort drawn from a Finnish population-based study. This imposes that all results and findings refer to men; this limitation should be acknowledged and discussed.
The sentence "Biomarkers should have sufficient specificity and sensitivity to be used in clinical practice" should be removed. A biomarker must be specific and sensitive to detect a given condition. Please revise the definition of "biomarker".
Controls were matched for age,BMI, SBP. Authors then adjusted the linear regression models to age, BMI, SBP and smoking; 3 out of 4 variables were matched. This is does not make sense. Models should be adjusted to other demographic variables that are not balanced between groups, which could explain some of the differences in metabolite levels. The most critical is the use of statin medication in AAA and controls; statin use can explain the difference in lipids metabolites found in this study. Authors must recalculate models with adequate adjustments.
Authors should explain the use of the statistical threshold, which sounds arbitrary.
Authors must avoid stating that these findings indicate "risk of AAA", the statical approach employed does not support this statement. Authors should revised this throughout the manuscript.
Do authors have any demographic or lifestyle information of these participants? These would improve the models and interpretability of the results. Without so, authors are unable to conclude that exposure to environmental toxins is associated with AAA in this cohort (line 182).
Line 184, authors say that 5 metabolites were associated with smoking. The results supporting this statement are not clearly presented.
Author Response
The manuscript submitted by Fernandes Silva et al presents an interesting and relevant study of metabolomics, genetic and environmental factors in patients with Abdominal Aortic Aneurysm (AAA), a life-threatening condition. The research topic is very relevant, but few topics need to be clarified.
- For this study the authors used a subset of a male cohort drawn from a Finnish population-based study. This imposes that all results and findings refer to men; this limitation should be acknowledged and discussed.
Reply: We agree that our study has limitations because it includes only Finnish men. We have now added this limitation on line 281: "The limitations of our study are that it included only middle-aged and elderly Finnish men, which limits the generalizability of our findings to women and other populations. Additionally, the use of cross-sectional data restricts our ability to establish causality between metabolic factors and AAA. "
- The sentence "Biomarkers should have sufficient specificity and sensitivity to be used in clinical practice" should be removed. A biomarker must be specific and sensitive to detect a given condition. Please revise the definition of "biomarker".
Reply; We agree with Reviewer 2. We have now revised our definition as suggested by the Reviewer 2 as follows on line 63: "A biomarker is a measurable indicator of a biological state or condition that is used to assess health or disease status, or to evaluate responses to a therapeutic intervention."
- Controls were matched for age, BMI, SBP. Authors then adjusted the linear regression models to age, BMI, SBP and smoking; 3 out of 4 variables were matched. This is does not make sense. Models should be adjusted to other demographic variables that are not balanced between groups, which could explain some of the differences in metabolite levels. The most critical is the use of statin medication in AAA and controls; statin use can explain the difference in lipids metabolites found in this study. Authors must recalculate models with adequate adjustments.
Reply: We have now recalculated the results. We have dropped out the matched for age, BMI, SBP. Reviewer 2 suggested the adjustment for statin treatment. We give now our data without adjustments and adjusted for statin use (p values).
- Authors should explain the use of the statistical threshold, which sounds arbitrary.
Reply: The statistical significance threshold was set at p≤ 5.0x10-5, based on a Bonferroni correction, which is a well-known and conservative method for delineating statistical thresholds. Bonferroni correction accounts for the number of tests performed (0.05 divided by the number of metabolites which was 1009 metabolites). This rigorous approach minimizes the potential for type I errors, ensuring that the reported associations are both statistically robust and likely to be biologically relevant.
- Authors must avoid stating that these findings indicate "risk of AAA", the statical approach employed does not support this statement. Authors should revised this throughout the manuscript.
Reply: We agree with Reviewer 2 and do not use the term "risk of AAA" in our manuscript.
- Do authors have any demographic or lifestyle information of these participants? These would improve the models and interpretability of the results. Without so, authors are unable to conclude that exposure to environmental toxins is associated with AAA in this cohort (line 182).
Reply: We have important information about lifestyle interpretability in our results. Specifically, we collected information on smoking status among our participants. The percentage of smokers was higher among the cases than in the controls, which provides a valuable lifestyle indicator relevant to our analysis of environmental toxins and their association with abdominal aortic aneurysm (AAA). Additionally, we have conducted comprehensive metabolomics profiling of all participants. Metabolomics offers profound insights into the biochemical processes occurring in the body. The plasma metabolome, which we analyzed, is strongly associated with body composition and lifestyle habits, thereby providing a significant connection between our participants' metabolic profiles and their environmental exposures as well as lifestyle choices. These data point are crucial as they not only help to control for key confounding variables but also augment the robustness of our models. Thus, even though direct evidence linking environmental toxins to AAA requires further investigation, the integration of smoking data and metabolomics insights allows us to draw more conclusions about the potential relationships among lifestyle factors, environmental exposures, and the development of AAA in this cohort.
We have revised the sentence in our manuscript - line 186: "Our findings indicate that exposure to environmental toxins, including PAHs, pesticides, and herbicides play a major role in the development of AAA [19, 36,37]."
- Line 184, authors say that 5 metabolites were associated with smoking. The results supporting this statement are not clearly presented.
We found that these 5 metabolites (2-naphthol sulfate, methylnaphthyl sulfate, 4-vinylphenol sulfate, 4-ethyl phenyl sulfate) were positively associated with AAA in our cohort before adjustment for statin use. After adjustment for statin use, 3 metabolites remained statistically significant associated with AAA. These associations were previously reported by Lorenz et al. (2021), who documented these associations in a longitudinal cohort study of HIV-positive and negative adults, thereby reinforcing the validity of these metabolites as indicators of tobacco exposure in diverse populations [38].
Reply: We improved our sentence for clarity on line 189: "Three metabolites positively associated with AAA in our study have been previ-ously associated with tobacco smoking (2-naphthol sulfate, methylnaphthyl sulfate and 4-vinylphenol sulfate) [38]."
